# Comparative efficacy and acceptability of antiepileptic drugs for classical trigeminal neuralgia: a Bayesian network meta-analysis protocol

Zongshi Qin,[1,2] Shang Xie,[3] Zhi Mao,[4] Yan Liu,[5] Jiani Wu,[1] Toshi A. Furukawa,[6] Joey S.W. Kwong,[7] Jinhui Tian,[8] Zhishun Liu[1]

For numbered affiliations see end of article.

**Correspondence to**
Professor Zhishun Liu;
liuzhishun@aliyun.com

## ABSTRACT

**Introduction** Trigeminal neuralgia (TN) affects 4 to 28.9/100 000 people worldwide, and antiepileptic drugs such as carbamazepine and oxcarbazepine are the firstline treatment options. However, the efficacy and safety of other antiepileptic drugs remain unclear due to insufficient direct comparisons.

**Objective** To compare the efficacy and acceptability of all currently available antiepileptic agents for the treatment of patients with classical TN.

**Methods** We will search the PubMed, EMBASE, Cochrane Library and Web of Science databases for unpublished or undergoing research listed in registry platforms. We will include all randomised controlled trials comparing two different antiepileptic drugs or one antiepileptic drug with placebo in patients with classical TN. The primary outcomes will be the proportion of responders and the number of subjects who dropout during the treatment. The secondary outcomes will include the two primary outcomes but in the follow-up period, changes in the self-reporting assessment scale for neuralgia and quality of life assessment. In terms of network meta-analysis, we will fit our model to a Bayesian framework using the JAGS and pcnetmeta packages of the R project.

**Ethics and dissemination** This protocol will not disseminate any private patient data. The results of this review will be disseminated through peer reviewed publication.

**PROSPERO registration number** CRD42016048640.

## Strengths and limitations of this study

► To the best of the authors' knowledge, this study will be the first network meta-analysis to assess the comparative efficacy and acceptability of all the available antiepileptic drugs for the treatment of classical trigeminal neuralgia.
► This study will be performed by Bayesian framework, which will enable us to estimate the probability of each intervention to be the best for each outcome.
► Owing to language barriers, the number of included trials may be potentially limited.

## INTRODUCTION

Classical trigeminal neuralgia (TN), a chronic pain disorder described as one of the most severe pains one can suffer, is characterised by paroxysms of unilateral, electric shock-like severe pain along the trigeminal nerve divisions.[1 2] It affects lifestyle because it can be triggered by common activities, such as eating, talking, shaving or brushing your teeth. The wind, chewing and talking also aggravate the condition in many patients.[2] It is estimated that approximately 4 to 28.9 per 100 000 people worldwide suffer from TN, and the number affected tends to be higher among women at all ages and even increases with age.[3 4]

At present, the cause of TN remains unclear.[5 6] One hypotheses is that the trigeminal nerve becomes compressed at the root entry zone by cerebral vessels.[7] Owing to the contradictory aetiology and poorly understood pathophysiological mechanisms underlying TN, a variety of therapeutic and surgical approaches have been developed to alleviate the associated pain and improve the quality of life for patients with classical TN.[8–10] Although many patients have obtained excellent outcomes from surgery, many others do not experience any pain relief.[11 12] Furthermore, the currently available surgical procedures are associated with various complications, particularly sensory loss in the trigeminal nerve territory, anaesthesia dolorosa and, rarely, ipsilateral hearing loss, depending on the technique.[13 14]

Hence pharmacological measures to improve clinical outcomes are needed. The most commonly used option is antiepileptic drugs, with phenytoin being the first drug to be used for classical TN with a positive effect.[15] Carbamazepine can reduce both the frequency and intensity of painful paroxysms and was first introduced by the US Food and

Drug Administration; however, its efficacy is compromised by poor tolerability.[16] Oxcarbazepine, a derivative of carbamazepine, is often used as an initial treatment for classical TN and has more favourable properties than carbamazepine related to its increased efficacy in epilepsy, greater tolerability and decreased potential for drug interactions.[17] Lamotrigine has also been reported as an effective add-on therapy,[18] whereas there is little evidence that other antiepileptic drugs, such as clonazepam, gabapentin, pregabalin and valproate, have a beneficial effect.[19–22] However, many of the studies are old with limited methodology, and were assessed as having low Grading of Recommendations Assessment, Development and Evaluation (GRADE) scores.[23]

To date, several systematic reviews have investigated the comparative efficacy and safety of antiepileptic drugs.[20 24–28] However, previous systematic reviews have only considered pairwise evidence from head to head comparisons and have thus failed to assess the comparative efficacy and acceptability of all the available antiepileptic drugs. Thus it is difficult to determine the best treatments for relieving pain with minimal adverse effects. In the present study, we choose a group of nine antiepileptic drugs, looking at drugs which were licensed for neuralgia in many countries and which were frequently used in clinical practice. We will apply network meta-analysis to integrate direct and indirect comparisons,[29 30] which could be used not only to strengthen inferences concerning the efficacy and acceptability of treatments but also to rank the efficacy and acceptability of antiepileptic drugs accordingly.[31]

The objectives of this systematic review and network meta-analysis are: (1) to compare all currently available antiepileptic drugs in terms of efficacy and acceptability in the treatment of classical TN; and (2) to determine which drug achieves the best balance between efficacy and adverse effects. The results of this study will augment findings based on current pairwise meta-analyses and are expected to provide important information to support clinical practice and health policy decisions.

## METHODS
This protocol will be conducted in accordance with the reporting guidance provided in the Preferred Reporting Items for Systematic Reviews and Meta-Analysis Protocols (PRISMA-P) statement and Checklist of Items to Include When Reporting a Systematic Review Involving a Network Meta-analysis.[32 33] The protocol is registered in PROSPERO (CRD: 42016048640). This study will not involve any private patient data; ethics approval was waived (see online supplementary file 1 for PRISMA-P checklist).

### Eligibility criteria
#### Study types
We will include randomised controlled trials (RCTs) comparing one antiepileptic drug with another antiepileptic drug as monotherapy or placebo for the treatment of TN. Quasi-randomised controlled trails allocating participants according to birth date or the consequences of enrolment will be excluded. The minimum duration for RCT inclusion will be set at 4 weeks. Trials with more than a two arm design will be considered only if the available data meet the criteria for an intervention. For trials with a crossover design, data will only be extracted from the first randomisation period.

### Participant characteristics
Only trials that enrolled participants with a diagnosis of classical TN according to standardised criteria, such as the International Headache Society's classification, International Classification of Headache Disorders, will be sought.[1 34] For studies using other extensive criteria for the diagnosis of classical TN, detailed diagnostic criteria must be reported (such as history or characteristics that have been confirmed by CT or MRI).[35] Studies examining symptomatic TN patients will not be included. Participants with comorbid conditions, such as anxiety, depression, epilepsy or other medical conditions, will not be eligible for inclusion. No limitations will be imposed on age, sex or nationality.

### Intervention types
We plan to include the following antiepileptic drugs: carbamazepine, lamotrigine, clonazepam, phenytoin, valproate, gabapentin, pregabalin, oxcarbazepine and topiramate. In addition to these antiepileptic drugs, we will also obtain information about interventions of interest from either pairwise RCTs or placebo controlled trails, as some RCTs design a placebo controlled arm as the comparator. Figure 1 illustrates the network plot of all possible direct comparisons between the eligible interventions.

### Outcome measures
Studies reporting one of the following will be included.

#### Primary outcomes
The primary objective of this review is to assess the efficacy and acceptability of antiepileptic drugs for classical TN; therefore, the following two outcomes will be used as the primary outcomes.
1. The proportion of responders to a self-reporting assessment scale for neuralgia. A responder was defined as a subject who obtained ≥50% reduction in pain score from baseline to the study endpoint (4–12 weeks) or a subject who obtained a pain reducing score of no less than the minimal clinically important difference. Pain scores will be extracted based on the visual analogue score, numerical rating score or any other validated scale for the assessment of overall TN symptoms when available.[36]
2. Treatment acceptability is defined as the proportion of patients who have intervention related adverse events during the 4–12 weeks.

*Secondary outcomes*

1. The proportion of responders with ≥50% pain reduction on a self-reporting assessment scale for neuralgia from baseline to the endpoint after follow-up.
2. The change in pain symptoms of TN from baseline to the endpoint (4–12 weeks), based on the visual analogue score, numerical rating score or any other validated scale for the assessment of overall TN symptoms when available.
3. The change in pain symptoms of TN from baseline to the endpoint after follow-up.
4. The quality of life based on measurement with a validated scale, such as the Short Form 36 Health Survey questionnaire.[37]

### Search strategy

We will identify RCTs through a comprehensive, systematic literature search, primarily utilising the PubMed, EMBASE, Cochrane Library and Web of Science databases. As publication bias caused by insufficient unpublished data can significantly bias the comparative efficacy results of network meta-analyses and modify rankings, we will also perform searches for unpublished or ongoing trials using the System for information on Grey Literature in Europe (SIGLE) as well as other registry platforms, such as Clinicaltrials.gov and the International Clinical Trials Registry Platform. Prior to completing this review, we will perform an additional search of each database and registration platform to guarantee that the most recent studies are included. We will use medical subject headings and text words related to 'trigeminal neuralgia' and 'randomised controlled trial' for the literature search. In addition, the reference lists of previous systematic reviews will be examined to ensure the quantity and accuracy of the included studies. The search strategy will be developed by JT and ZL; we anticipate that the databases will be searched from their inception to 30 September 2017 (see online supplementary file 2 for the search strategies for PubMed, EMBASE and Cochrane Library).

### Data collection process

Two authors (SX and ZM) will scan the titles and abstracts of the trials after duplicated records have been excluded using EndNote X7 (Thomson Reuters, New York, New York, USA). The scanning will be performed using EndNote, and all trials will be allocated to the following five groups: inclusion group, non-patient group, intervention group, outcome group and awaiting group. A prior data collection process will be conducted using an electro table created with Excel software, which has been used in our previous study.[38] The table will consist of four sheets, including general information (author list, publication year and journal), characteristics of the included trials (diagnostic criteria, age range, study drugs and dose range), risk of bias assessed using the Cochrane risk of bias tool and outcome data extraction (number of participants who responded to treatment and the number who dropped out during the treatment). All original data will be submitted as an attachment. A flowchart illustrating this design is presented in figure 2.

### Quality assessment

Two authors (JW and YL) will use the Cochrane risk of bias tool to assess the risk of bias of the eligible studies,

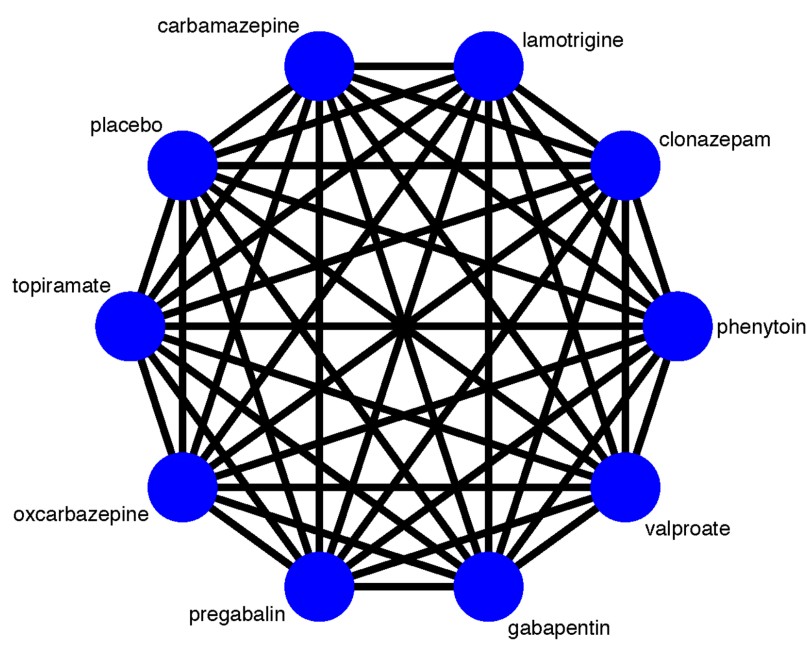

**Figure 1** Network plot of all possible direct comparisons between the eligible interventions.

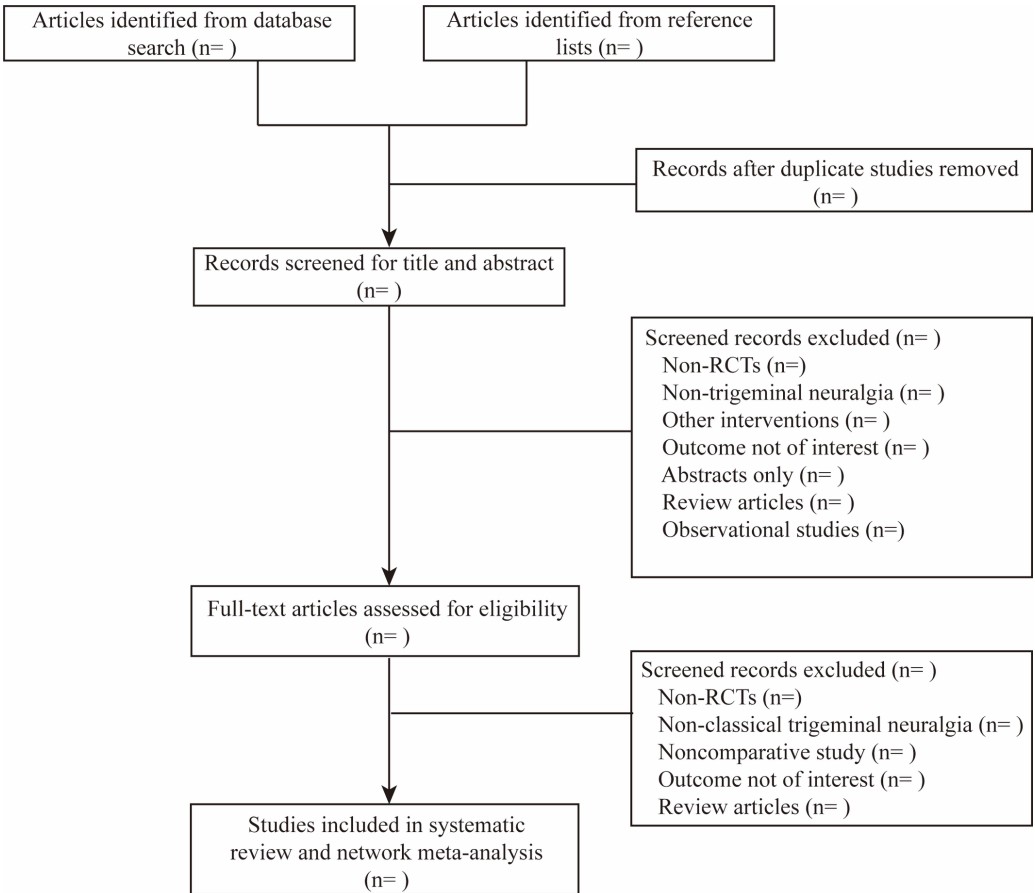

**Figure 2** PRISMA flowchart. RCT, randomised controlled trial.

covering randomisation, concealment allocation, blinding and other biases.[39] As inadequate concealment could potentially fail the randomisation test, two independent review authors will pay particular attention to the adequacy of random allocation concealment and blinding. The other sources of bias will be assessed considering the sample size calculation method, diagnostic criteria, reporting of withdrawals and follow-up. Two authors (JSWK and JT) will assess the quality of evidence using the GRADE framework, covering study limitations, inconsistency, indirectness, imprecision and publication bias.[40] The methods for rating the quality of direct comparisons are the same as the methods used in traditional meta-analyses, and the following steps will be used in the whole assessment procedure: (1) presenting direct and indirect effect estimates; (2) rating the quality of direct and indirect estimates; (3) presenting the results of the network meta-analysis; and (4) rating the quality of the network meta-analysis effect estimates.

### Dealing with missing data

To obtain missing data, we will initially contact the senior or corresponding author. If no one responds, we will estimate the missing data as follows. For studies failing to report the number of responding patients after treatment, instead of providing the mean and SD, we will calculate the number of responding patients employing a validated imputation method.[41] In addition, we will also estimate missing data from graphs when possible. For trials that cannot be extracted or estimated, the available data will be excluded, and the reason for exclusion will be reported.

### Statistical analysis

The method used for data synthesis will be based on mixed treatment meta-analysis. To examine comparisons, we will use Stata (13.0; Stata Corporation, College Station, Texas, USA) to synthesise data and will present the comparison results if the included studies are sufficient for each pairwise comparison. We will use a random effects model to combine the data, and the outcomes of continuous and binary variables will be presented as standardised mean differences (SMDs) and ORs with 95% CIs. For indirect comparisons, we will perform an arm based network meta-analysis for all treatments using a random effects model with a Bayesian framework using the pcnetmeta package of the R project, which can conduct calculations using JAGS software.[42–44] This will enable us to estimate the best probability of each intervention for each positive outcome, given the results of the multiple treatment meta-analysis. At least one network focusing on the response rate for pain relief will be constructed, in which a statistically significant difference defined as the null value will not

be included in the 95% CI. All models will be utilised for 50 000 simultaneous iterations based on the data and the description of the proposed distributions for relevant parameters, and the first 10 000 iterations will be discarded to avoid potential impact on the arbitrary value. For continuous outcomes and binary outcomes, the OR and SMD values will be presented with the 95% credible interval (CrI).

To describe relationships among different treatments, a network plot will be created to show direct comparisons between arms based on different outcomes.[42] In addition, the effectiveness of each treatment among all available treatments will be ranked by calculating the OR in order, and plots of the treatment rank probabilities will be generated to rank the various treatments for each outcome using the functions in package pcnet-meta.[42 43] We will also present a cluster rank table to synthesise the efficacy and acceptability of each drug (using two primary outcomes). The table will consist of two triangles: the upper right triangle will illustrate the acceptability and the lower left triangle will illustrate the efficacy.[31] For pairwise meta-analyses we will use Stata 13.0. For network meta-analyses we will use JAGS and R project.

### Assessment of heterogeneity

Heterogeneity, which plays a pivotal role in both standard meta-analyses and network meta-analyses, refers to the degree of disagreement between study specific treatment effects and constitutes the basis of inconsistency. To test the heterogeneity of each pairwise comparison, we will use the I² statistic.[45]

### Assessment of transitivity and similarity

In addition to the heterogeneity assessment using the I² statistic, the assumption of transitivity and similarity based on clinical and methodological characteristics will be assessed. It should be noted that it is difficult to identify these effect modifiers using statistical analysis. We will assume that intervention effects are transitive in this network meta-analysis because we will only focus on antiepileptic drugs, and we will investigate similarity based on clinical characteristics, such as antiepileptic drug dose, period of treatment and severity of pain symptoms at baseline, as well as according to methodological characteristics, such as study quality.[46] All of these effect modifiers will be judged and reported before the network meta-analysis is conducted.

### Assessment of inconsistency

Evaluation and explanation of inconsistency is another basic objective of a network meta-analysis. In this context, inconsistency refers to the degree of difference between direct and indirect comparisons and can be evaluated only when a loop exists in the evidence network. This means that inconsistency assessment using a design by treatment interaction model cannot be conducted if the structure of this network is a 'star network' (ie, all interventions have a single mutual comparator, such as a placebo).[47 48] For such cases, we will test inconsistency using a node splitting model.[49]

To identify inconsistency among the included trials of the network, we will use Stata, performing the Z test to compare direct and indirect summary effects in specific loops.[50] If there is no inconsistency between loops or designs, we will use a consistency model to calculate the data. For cases of significant incoherence, we will initially look for data extraction errors in loops that present inconsistency and in comparisons with large heterogeneity.[51] After the data have been scrutinised, we will investigate possible sources of inconsistency within the clinical and methodological variables suspected of being potential sources of either heterogeneity or incoherence in each comparison specific group of trials. If an important inconsistency cannot be explained, we will consider avoiding synthesis of the related network.

### Additional analyses

To ensure the quality of this review, studies not reporting blinding will be excluded prior to data synthesis because blinding plays a vital important role in the RCT. We will assess heterogeneity quantitatively using the I² statistic, and if an I² value is >50%, we will explore the source of heterogeneity. We will initially perform sensitivity analysis by excluding trials rated as having a high risk of bias. Additionally, meta-regression or subgroup analysis will be used to explore possible sources of heterogeneity if the number of included trials is sufficient. For network meta-regression, we will use a random effects network meta-regression model to examine potential factors.

## DISCUSSION

To the best of our knowledge, no network meta-analyses comparing the use of antiepileptic drugs for the treatment of classical TN have been conducted to date. Previous systematic reviews have compared only a single drug to other types of drug or therapy.[20 24–28] This makes it difficult to obtain a clear understanding of the effectiveness of the various different conservative treatments for this disorder. A network meta-analysis can be used to perform indirect comparisons and allows parameters for direct and indirect comparisons to be synthesised. To ensure the quantity and quality of the potentially included RCTs, we will perform an extensive literature search and predefine rigorous inclusion criteria. Also, we will assess the quality of evidence using the GRADE framework. Although a ranking of the included interventions will be generated, with the exception of findings, the quality of evidence should also be considered. We hope that the results of this review will help clinicians make more accurate treatment decisions and promote additional research into conservative treatments for classical TN.

## Amendments

If it is necessary we will update this protocol in the future. We will submit the original protocol, final protocol and summary of changes as a supplement.

**Author affiliations**

[1]Department of Acupuncture and Neurology, Guang'anmen Hospital, China Academy of Chinese Medical Sciences, Beijing, China

[2]School of Life Sciences, Beijing University of Chinese Medicine, Beijing, China

[3]Department of Oral and Maxillofacial Surgery, Peking University School and Hospital of Stomatology, Beijing, China

[4]Department of Critical Care Medicine, Chinese People's Liberation Army General Hospital, Beijing, China

[5]Data Centre of Traditional Chinese Medicine, China Academy of Chinese Medical Sciences, Beijing, China

[6]Department of Health Promotion and Human Behaviour, Kyoto University Graduate School of Medicine/School of Public Health, Kyoto, Japan

[7]JC School of Public Health and Primary Care, Faculty of Medicine, The Chinese University of Hong Kong, Hong Kong, Hong Kong, China

[8]Evidence Based Medicine Centre, Lanzhou University, Lanzhou, China

**Contributors** ZQ, SX and ZM conceived the study, and contributed equally to this study. JT and SX developed the search strategies. ZQ and SX wrote the first draft. TAF and ZL revised the draft. SX and ZM will independently screen potential studies and extract data from the included studies. JW, JSWK, JT and YL will assess the risk of bias and summarise the evidence. ZM, SX, JSWK and ZL will address the missing data, if any. ZQ and JT will perform the statistical analysis. ZL and JT will arbitrate in cases of disagreement and ensure the absence of errors. All authors approve the publication of this protocol.

**Funding** This research received no specific grant from any funding agency in the public, commercial or not-for-profit sectors.

**Competing interests** None declared.

**Patient consent** Parental/guardian consent obtained.

**Provenance and peer review** Not commissioned; externally peer reviewed.

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
