## [Reviewer comments · BMJ Open]

ARTICLE DETAILS

TITLE (PROVISIONAL)	Comparative Efficacy and Acceptability of Antiepileptic Drugs for Classical Trigeminal Neuralgia: A Bayesian Network Meta-Analysis Protocol
AUTHORS	Qin, Zongshi; Xie, Shang; Mao, Zhi; Liu, Yan; WU, Jiani; Furukawa, Toshi; Kwong, Joey; Tian, Jinhui; Liu, Zhishun

VERSION 1 – REVIEW

REVIEWER	Joanna Zakrzewska UCLH NHS Foundation Trust UK
REVIEW RETURNED	01-May-2017

GENERAL COMMENTS	This is a protocol for carrying out a novel systematic review of anti-epileptic drugs used in the management of classical TN and reported according to PRISM protocol. Introduction : TN is not most common neuralgia . It may be worth mentioning that many of the studies are very old, there is a lack of detail and their quality is poor, GRADE scores are low Clinical Evidence Zakrzewska Linskey 2014 Methods There is no PICO as such but characteristics of the studies are provided. Not entirely clear about variations on the outcome measures as all on pain relief- is is just a question of robustness of the measure. What is meant by change in pain symptom is this a global impression of change ? Acceptability in primary outcomes is only patients who drop out. Many patients will continue medications despite side effects because of severe pain.so any side effects should form part of the outcome measures as a separate measure. Concern that may get many Chinese articles that have not been through careful peer review. Funding nil but who is funding the publication, who is the sponsor, this is mentioned on the PROPERO website References 8 and 14 are the same and essentially 18 is the same group and same results. No reference is provided for oxcarbazepine. Use a more up to date reference instead of reference 2 e.g Zakrzewska Linskey BMJ
--

REVIEWER	Tuleasca, Constantin Lausanne University Hospital, Neurosurgery Service and Gamma Knife Center, Switzerland University of Lausanne, Faculty of Biology and Medicine
REVIEW RETURNED	19-May-2017

GENERAL COMMENTS	Interesting study, deals with a major problem regarding the anti epileptic drugs and their use in classical trigeminal neuralgia. Mainly due to the heterogeneity, very challenging protocol. If this adds to the current data in terms of safety, efficacy and toxicity, could be a plus in the pharmacological management.
--

REVIEWER	Jing Zhang University of Maryland, USA
REVIEW RETURNED	05-Jul-2017

GENERAL COMMENTS	This paper conducted a systematic review and network meta-analysis to compare the efficacy and acceptability of antiepileptic drugs for trigeminal neuralgia, which has not been done before. However, it is not clear what results were obtained and what conclusion were drawn. What are the assessment results of transitivity and similarity? It is not clear why and how the 9 antiepileptic drugs were selected in this project. For indirect comparisons, a random effects model network meta-analysis will be developed. What random effects model does this refer to? There are two broad categories of methods for network meta-analysis: contrast-based and arm-based. Have the author ever considered the arm-based method (Zhang 2014)? Zhang, J., Carlin, B.P., Neaton, J.D., Soon G.G., Nie L., Kane, R., Virnig B.A., and Chu, H. (2014). "Network meta-analysis of randomized clinical trials: Reporting the proper summaries". Clinical Trials, 11 (2): 246-262.
---

VERSION 1 – AUTHOR RESPONSE

Reviewer #1

Comment 1: Introduction TN is not most common neuralgia. It may be worth mentioning that many of the studies are very old, there is lack of detail and their quality is poor, GRADE scores are low Clinical Evidence Zakrzewska Linskey 2014.

Answer:

Many thanks for picking up this error. Accordingly, we have changed the sentence, as follows: "It is estimated that approximately 4 to 28.9 per 100,000 people worldwide suffer from TN, and the number affected tends to be higher among women at all ages and even increases with age." We've already quoted the following reference to indicate that "many of the studies are out-of-date with limited methodology, and were assessed as low GRADE scores" (Zakrzewska JM, Linskey ME. Trigeminal neuralgia. *BMJ Clin Evid*. 2014. pii:1207), please see page 3.

Comment 2: Methods There is no PICO as such but characteristics of the studies are provided. Not entirely clear about variations on the outcome measures as all on pain relief is just a question of robustness of the measure. What is meant by change in pain symptom? Is this a global impression of change?

Answer:

We have reported the information of PICO at the beginning of Methods section, under a subheading "Eligibility criteria", including 'Participant characteristics' (Classical TN), 'Intervention types' (antiepileptic drugs), and 'Outcome measures'. Please see page 5. The change in pain symptom of secondary outcome denotes continuous data of pain relief.

Comment 3: Methods Acceptability in primary outcomes is only patients who drop out. Many patients will continue medications despite side effects because of severe pain. So any side effects should form part of the outcome measures as a separate measure.

Answer:

Thanks for your professional suggestions. We've changed the acceptability in primary outcomes as follows: "Treatment acceptability is defined as the proportion of patients who have intervention related adverse events during the 4 to 12 weeks."

Comment 4: Methods Concern that may get many Chinese articles that have not been through careful peer review.

Answer:

We've also considered your concern, thus, Chinese databases will not be searched. However, studies that could be searched on the English databases will be scanned or included, if the studies could meet our inclusion criterias.

Comment 5: Funding Funding nil but who is funding the publication, who is the sponsor, this is mentioned on the PROSPERO website.

Answer:

This study has no sponsor and we have reported this at the end of the manuscript.

Comment 6: References References 8 and 14 are the same and essentially 18 is the same group and same results. No reference is provided for oxcarbazepine. Use a more up to date reference instead of reference 2 (e.g. Zakrzewska Linskey BMJ)

Answer:

Many thanks for picking up this error. We've already deleted the repeated reference, and adjusted the order accordingly. We've quoted the following reference in the Background section of the manuscript instead of reference 2 (Zakrzewska JM, Linskey ME. Trigeminal neuralgia. BMJ. 2015; doi: 10.1136/bmj.h1238). In addition, we've also added the following reference in the Background section for oxcarbazepine (Zakrzewska JM, Patsalos PN. Long-term cohort study comparing medical (oxcarbazepine) and surgical management of intractable trigeminal neuralgia. Pain. 2002;95:259-66).

Reviewer #2

Comment 1: Interesting study, deals with a major problem regarding the anti-epileptic drugs and their use in classical trigeminal neuralgia. Mainly due to the heterogeneity, very challenging protocol. If this adds to the current data in terms of safety, efficacy and toxicity, could be a plus in the pharmacological management.

Answer:

Many thanks for reviewing our manuscript. The results of this study will be produced in the next year or two (see also reply to first comment by Reviewer #3).

Reviewer #3

Comment 1: This paper conducted a systematic review and network meta-analysis to compare the efficacy and acceptability of antiepileptic drugs for trigeminal neuralgia, which has not been done before. However, it is not clear what results were obtained and what conclusion were drawn.

Answer:

This study will assess the comparative efficacy and acceptability of 9 antiepileptic drugs for the classical trigeminal neuralgia. We anticipate the findings of this study will be produced in the next year or two.

Comment 2: What are the assessment results of transitivity and similarity?

Answer:

The assumption of transitivity and similarity will be assessed mainly base on clinical and methodological characteristics. We will assume that intervention effects are transitive in this network meta-analysis because we only include antiepileptic drugs, and we will investigate similarity based on clinical characteristics, such as antiepileptic drug dose, period of treatment, and severity of pain symptoms at baseline, as well as according to methodological characteristics such as study quality. Please see the 'Assessment of transitivity and similarity' section in page 9.

Comment 3: It is not clear why and how the 9 antiepileptic drugs were selected in this project.

Answer:

We chose a group of 9 antiepileptic drugs looking at the drugs which were licensed for neuralgia in many countries and which were frequently used in clinical practice.

Comment 4: For indirect comparisons, a random effects model network meta-analysis will be developed. What random effects model does this refer to? There are two broad categories of methods for network meta-analysis: contrast-based and arm-based. Have the author ever considered the arm-based method (Zhang 2014)?

Answer:

Thanks for your professional suggestions. In the Statistical analysis section of manuscript, the model refer to arm-based parameterization random effects model. Accordingly, we've already quote the following reference (Zhang J et al. Network meta-analysis of randomized clinical trials: Reporting the proper summaries. *Clinical Trials*. 2014;11:246-262), please see page 8.

VERSION 2 – REVIEW

REVIEWER	Jing Zhang University of Maryland, USA
REVIEW RETURNED	12-Aug-2017

GENERAL COMMENTS	This seems a nice protocol. But its significance is weakened because it is only a protocol and no results are obtained yet. Though the authors claimed that they will use the arm-based method to do the analysis. It is not clear what exact models they will use. The 2014 Clinical Trials paper proposed models for binary outcomes. Other papers, for example, "Detecting outlying trials in network meta-analysis" and "A Bayesian missing data framework for generalized multiple outcome mixed treatment comparisons" may provide ideas for continuous outcomes and multiple outcomes. It is not clear why the authors want to conduct analysis using both winbugs and STATA. Do the authors expect inconsistency of the results between these two softwares? There are some existing R packages that the authors may want to consider.
---

VERSION 2 – AUTHOR RESPONSE

Comments from Reviewer #3

Comment 1: Though the authors claimed that they will use the arm-based method to do the analysis. It is not clear what exact models they will use. The 2014 Clinical Trials paper proposed models for binary outcomes. Other papers, for example, "Detecting outlying trials in network meta-analysis" and "A Bayesian missing data framework for generalized multiple outcome mixed treatment comparisons" may provide ideas for continuous outcomes and multiple outcomes.

Answer 1: We will use Chaimani model for network meta-analysis, which could either calculate the continuous outcomes and binary outcomes. We have added the information in the statistical analysis section: ' For indirect comparisons, network meta-analysis will be developed in a Bayesian framework using Markov chain Monte Carlo simulation methods in WinBUGS (Medical Research Council's Biostatistics Unit, Cambridge, UK) with a Chaimani model.'

Comment 2: It is not clear why the authors want to conduct analysis using both winbugs and STATA. Do the authors expect inconsistency of the results between these two softwares? There are some existing R packages that the authors may want to consider.

Answer 2: We use winbugs software to calculate the data, after that, we will use Stata software to draw the pictures. We have clarified the role of Stata software in Statistical analysis section: 'The effectiveness of each treatment among all available treatments will be ranked by calculating the OR in order, and plots of the surfaces under the cumulative ranking curves (SUCRAs) will be generated to rank the various treatments for each outcome using Stata software.' Please see reference 44.

VERSION 3 – REVIEW

REVIEWER	Jing Zhang University of Maryland
REVIEW RETURNED	14-Sep-2017

GENERAL COMMENTS	The quality of writing still needs improvement. There are sentences with grammar errors. Some sentences are not clear. The authors need to double check the cited references. For example, on p.8 lines 28-34. For the Chaimani model, the authors cited reference 42. However, 42 "Lu G, Ades AE. Combination of direct and indirect evidence in mixed treatment comparisons. Stat Med. 2004;23:3105–3124" is not a paper of Chaimani. It is not clear why the model in this paper is called a Chaimani model. It is better if the authors could provide the Chaimani model. It is great that the arm-based parameterization will be used as it provides more comprehensive summaries. However, it is not clear how the Chaimani model can produce the arm-based summaries. The paper "Performing arm-based network meta-analysis in R with the pnetmeta package. Journal of Statistical Software. 80 (5): doi: 10.18637/jss.v080.i05" may be helpful to produce these summaries. But it was an R package instead of STATA. In addition, the authors seems to be confused about the arm-based and contrast-based models. This will need more efforts and the statistical analysis part needs improvement. Refer to the following papers for more information. "Absolute or relative effects? Arm-based synthesis of trial data". Research Synthesis Methods. DOI: 10.1002/jrsm.1184. "Rejoinder to the Discussion of 'A Bayesian missing data framework for generalized multiple outcome mixed treatment comparisons' by S. Dias and A.E. Ades". Research Synthesis Methods, 7 (1): 29-33.
---

VERSION 3 – AUTHOR RESPONSE

Comments from Reviewer #3

Comment 1: The quality of writing still needs improvement. There are sentences with grammar errors. Some sentences are not clear.

Answer 1: Thanks for your suggestion. We have asked a colleague who got his PhD degree in the United Kingdom to polish the manuscript's language.

Comment 2: The authors need to double check the cited references. For example, on p.8 lines 28-34. For the Chaimani model, the authors cited reference 42. However, 42 "Lu G, Ades AE. Combination of direct and indirect evidence in mixed treatment comparisons. Stat Med. 2004;23:3105–3124" is not a paper of Chaimani. It is not clear why the model in this paper is called a Chaimani model. It is better if the authors could provide the Chaimani model.

Answer 2: The Chaimani model was reported by Prof. Anna Chaimani, and the description of this model could be found at following websites:

<http://www.mtm.uoi.gr/images/3.binarymodeldescription.pdf>

<http://www.mtm.uoi.gr/images/3.continuousmodeldescription.pdf>

However, we would like to use JAGS and R project instead of winbugs for network meta-analysis in this study. The reason has been described in answer 2.

Comment 3: It is great that the arm-based parameterization will be used as it provides more comprehensive summaries. However, it is not clear how the Chaimani model can produce the arm-based summaries. The paper "Performing arm-based network meta-analysis in R with the pcnetmeta package. Journal of Statistical Software. 80 (5): doi: 10.18637/jss.v080.i05" may be helpful to produce these summaries. But it was an R package instead of STATA. In addition, the authors seems to be confused about the arm-based and contrast-based models. This will need more efforts and the statistical analysis part needs improvement. Refer to the following papers for more information.

"Absolute or relative effects? Arm-based synthesis of trial data". Research Synthesis Methods. DOI: 10.1002/jrsm.1184.

"Rejoinder to the Discussion of 'A Bayesian missing data framework for generalized multiple outcome mixed treatment comparisons' by S. Dias and A.E. Ades". Research Synthesis Methods, 7 (1): 29-33.

Answer 3: Thanks for your professional suggestion. We initially planned to conduct network meta-analysis using Stata and winbugs given that the team members have been familiar with aforementioned software and Chaimani model. In the past weeks, we have tested the function of pcnetmeta package with previously published dataset, and scanned the related research publication. The pcnetmeta package in R is based on Bayesian theory, which combines the strength computing function of JAGS software and the special data integration and powerful graph drawing function of R project. At the same time, this package can draw many kinds of plots. Thus, we decided to use R and JAGS, which greatly meets actual needs of us to deal with complicated network meta-analysis, and can perform the analysis with minimal computational effort.

Accordingly, we have revised the related statistical analysis section as following:

"For indirect comparisons, we will perform arm-based network meta-analysis for all treatments using a random effects model with a Bayesian framework using the pcnetmeta package of R project, which could conduct calculation by calling JAGS software.42-44"

"To describe relationships among different treatments, a network plot will be created to show direct comparisons between arms based on different outcomes.42 In addition, the effectiveness of each treatment among all available treatments will be ranked by calculating the OR in order, and plots of the treatment rank probabilities will be generated to rank the various treatments for each outcome using the functions in package pcnetmeta.42 43"

"For pair-wise meta-analyses we will use Stata 13.0. For network meta-analyses we will use JAGS and R project.42 43"

Reference

42. Lin L, Zhang J, Hodges J, Chu H. Performing arm-based network meta-analysis in R with the pcnetmeta package. Journal of Statistical Software. 2017;doi: 10.18637/jss.v080.i05

43. Lin L, Zhang J, Chu H. pcnetmeta: Methods for patient-centered network meta-analysis. 2016. R package version 2.4 <http://CRAN.R-project.org/package=pcnetmeta>

44. Zhang J, Carlin B.P, Neaton J.D, et al. Network meta-analysis of randomized clinical trials: Reporting the proper summaries. Clinical Trials. 2014;11:246-262

VERSION 4 – REVIEW

REVIEWER	Jing Zhang University of Maryland, USA
REVIEW RETURNED	08-Nov-2017
GENERAL COMMENTS	All previous comments are addressed well and I don't have further comments.